# Teaching Marketing Research at the University Level—From Academic and Professional Perspectives

Javier de la Ballina * and Silvia Cachero

School of Economics and Business, University of Oviedo, 33006 Oviedo, Spain
* Correspondence: fballina@uniovi.es; Tel.: +34-985103918

**Abstract:** The evolution of marketing research (MR) has run parallel to marketing; however, nowadays, statistical techniques and data technologies are gaining more importance. The need for alignment regarding MR training between professors, students, and professionals is becoming increasingly urgent. This work continues on a double survey administered to professors and MR professionals in Spain to determine their proposals for adapting MR courses' format and content to the companies' current information needs. The results show that not only do professionals lead in terms of wanting changes to training, but also that these professionals are by no means extreme in their demands regarding university professors. The findings also show that, although there are significant differences in priorities at both the statistical and technical levels, the solution may be to combine and slightly adjust the current mandatory MR courses in business administration degrees. In addition, an elective course that develops training in new data and intelligent technologies for MR should be implemented.

**Keywords:** marketing research; higher education; evolution; university subject; professors; professionals; data; technology

## 1. Introduction

Companies are increasingly seeking information for decision-making in today's rapidly changing world. The importance of training in market research (MR) runs in parallel with companies' needs.

Current technological innovation has produced significant changes in the MR field, which is committed to digital, innovative, and transformative data processes or methodologies. These changes may affect the content, tools, and format of MR courses at the university level.

Firstly, statistical engineering has allowed for the development of software that can apply increasingly more complex techniques to both SQL (Structured Query Language)- and NoSQL (non-relationally stored)-managed data [1].

New working technologies that are also are incorporated into this field in terms of both data collection and analysis, as well as data storage, include: big data, extract-treat-load (ETL), data warehouses [2], apps, and smartphones [3]. Moreover, it is essential to highlight the importance of neuromarketing, which has critical applications in research with virtual reality and artificial intelligence [4].

Furthermore, the abundance of microdata on all people has brought the critical question of ethics in MR and the management of social networks and bots to the table [5,6].

The traditional dichotomy between university training in MR and professional practice must adapt to this innovation context. The literature offers some studies incorporating new statistical or digital techniques. However, only some works provide a global vision of the convergence of both approaches [7].

This paper aims to provide empirical evidence for the reflections of Malhotra and Peterson [8]. They reflected on the influence of new smart technologies on MR research

methodologies. However, they should have specified them and addressed the issue of necessary university training to use the new MR methodologies. Therefore, this work conducts a comparative study between university professors and professionals from large MR companies to determine the incorporation of new statistical and digital technologies into MR courses offered at the university level [6]. More specifically, this study also examines preexisting points in programs that may reduce or remove [9] the need to facilitate the adaptive change of MR.

This study has the following structure: first, it examines the literature related to the conceptual evolution of MR, emphasizing the new perspectives of "Intelligence" versus "Research". In addition, it reviews the literature related to teaching MR at the university level with a view to the perspectives of professors, students, and professionals. Second, it presents the methodology, the hypotheses for the empirical work, and the results of the statistical analysis of the data obtained through surveys administered to the professionals and professors. Finally, this article ends with the presentation of the study's conclusions and a discussion of their practical implications.

## 2. Literature Review

### 2.1. About the Evolution of the Concept of Marketing Research

The evolution of MR throughout history may be organized into a series of main stages. First, there is the initial stage, which consisted of the formal emergence of MR in 1910 with the creation of the first MR department in Curtis Publishing Company. A few years later, a university department was created in the Harvard Business School, thus beginning the training of students in MR [10].

The period between 1920 and 1940 represents an acceleration stage [11], with the term MR becoming more popular in the business and academic spheres with the relevance of sampling and questionnaire analysis techniques. In addition, telephone surveys were developed in this period, and the first panels appeared.

Between 1940 and 1960, the phenomenon of the 4Ps influenced the MR consolidation stage, which affected operations in MR and helped it become a fundamental activity for managers and decision-making. This decade also bore witness to the US government creating its first statistical offices that regularly gathered demographic, social, and economic data [10].

The 1960s and 1970s represent the beginning of the behavioral research stage [11]. This moment became necessary as the need to divide consumers into groups and segment the market arose in the marketing field. This stage developed consumer motivation studies using tools such as in-depth interviews and focus groups. This period promoted multivariate classification statistical techniques known as cluster analysis.

The office automation era began when the first computers and statistical packages were launched on the market [11]. Quantitative research began to take precedence. These technological advances allowed the use of attitude scales, factor and discriminant analysis, mathematical models, and simulation implementation. Researchers began first to introduce spreadsheets (first with Lotus, and later with Excel), thus generalizing the use of statistical software (from BMDP to IBM's SPSS) [12].

In the 2000s, how managed information was managed underwent a sudden and dramatic change with the development of the Internet [12], giving rise to the technological revolution stage. The widespread introduction of the Internet within the field of MR multiplied the options for accessing all types of data and carrying out market studies over time, which had previously been quite tricky.

This stage also saw the appearance of CRM (Customer Relationship Management), a database storing all interactions between a company and its customers [7]. This element led to sharing and maximizing the information available about a customer. All possible information about a given customer could be collected and accessible on any company device, allowing for new objectives regarding customer satisfaction and loyalty [5].

In the late 2010s, the rapid development of smart technologies began [13]. This change consisted of a new technological revolution that used smart intelligence as its basis to make both resources and media act and respond intelligently, learn from experiences, and be capable of solving previously unseen situations thanks to their data processing capabilities and interactions with the environment. Among these technologies, the following are noteworthy: big data, machine learning, blockchain, artificial intelligence, and, undoubtedly, nanotechnology [14].

The MR process changed with the development of digital transformation, which supported a transition toward new concepts such as business intelligence (BI), competitive intelligence (CI), and market intelligence (MI).

The development of smart technologies entailed perfecting and better-integrating data within organizations [7], which translates into better business information. Now appears business intelligence. BI integrates all the techniques based on computer or technological means to detect, deepen, and analyze data related to a company's activities to improve decision-making [10]. It is a decision support system that aims to enhance and optimize organizational performance and decision-making (see Table 1).

**Table 1.** Characteristics of business intelligence.

| BI System | Description |
| --- | --- |
| Decision support system | The manager has essential information at the right time to use in strategic and operational decision-making. |
| Technology-based | It works with technology that allows obtaining quality data, processing it, and then presenting it appropriately (ELT, data mining, data warehouse, control panel). |
| Complete | It covers the process of collecting, processing, analyzing, and distributing business information. |
| Open concept | It is not a specific technology but a set of variable intelligent elements. |

It is essential to note the differences between BI and CI. All information within the organization is the domain of BI, but information about current competitors is the domain of CI. Gibbons and Prescott [15] note that CI refers to the processing of obtaining, analyzing, interpreting, and disseminating information of strategic value about the industry and competitors, which is transmitted to decision-makers as soon as possible. CI allows organizations to make decisions to get ahead of their competitors [12].

BI has a broader perspective than CI, as it focuses on events beyond current competitors and the primary industry and attempts to predict and anticipate significant changes that may arise, both at the specific sector and overall market levels.

The second relevant term is MI. It seeks to collect all types of data that provide the company with a general and global vision of the market, not only of the current markets, such as in the case of CI.

The Academy of Market Intelligence (AMI) established [16] that MI seeks to obtain relevant results about a company's market through internal and competitive analyses supported with information. From the professional perspective, QuestiónPro [17] considers MI to be the information or data that an organization obtains from the industry in which it operates to determine its existing segmentation, penetration, opportunities, and metrics. MI is a strategy that uses internal and external data to supervise the market continually, rather than ad hoc, to improve competitiveness.

### 2.2. Teaching Marketing Research at the University Level

MR has always gone together with marketing, meaning that their implementation at the university level is connected. In the US, the presence of marketing at the university level dates to the beginning of the 20th century, supported by commercial investigation, which strongly focuses on sales. In Europe, and more specifically in Spain, the delay was considerable, with the discipline arriving several decades later [12].

There are five fundamental periods in the evolution of MR at the university level in the case of Spain [18,19]:

- A prefoundational period that runs parallel to the appearance and development of marketing. It can be dated to the first decade of the 20th century, specifically 1911, when the Chamber of Commerce in Barcelona organized the first marketing course in Spain, which included aspects of MR. This period was long and practically lasted until 1940 [20].
- A transitory period characterized by the creation of political, economic, and commercial sciences faculties at the general level, starting in 1943. However, these faculties scarcely dedicated much attention to marketing and, consequently, to MR [21]. They were much more focused on training in economic disciplines.
- A foundational period in the 1960s and 1970s with the arrival of the business economics approach to Spanish universities. Marketing was independent; however, there was still a minimal number of courses, disparate contents and denominations, and few compulsory courses [18]. The term commercial investigation maintains its use to refer to lessons or classes related to information in marketing.
- A recognition period starting in the 1980s and 1990s with the functional advancement of Business Administration (BA) degrees, which distinguished between training in finances, organization and administration, personnel management, and marketing [22]. For example, in Spain in the 1990s, the number of marketing professors went from 27 to 206 [23].
- The period of MR development from the 21st century onwards. The course became a core, compulsory course in BA degrees, with the emergence of the first master's and bachelor's degrees dedicated explicitly to MR [19].

Bridges [21] was the first academic to take an interest in the need for change and adaptation in the teaching of MR. The fieldwork Bridges conducted with students began by detailing the students' difficulty with MR, despite them also considering a relevant course for their training. Bridges also found that interactive activities and group work caught students' attention more than conferences or independent work.

The author also reviewed the leading textbooks on MR and concluded that, despite the advances and changes in IT, the content primarily covered had stayed the same: use of secondary data, questionnaire design, sampling, data analysis, and presentation of results.

There is a debate between those who recommend dedicating more time to developing quantitative skills and those who choose to focus on qualitative skills. Those that make up the first group believe that there should be more focus on working with data and that data analysis is better to help managers to understand business problems [24]. However, those who belong to the other school of thought believe that numerical databases cannot explain consumer behavior, resulting in the increased importance of qualitative research [25].

Bridges [23] developed a later study based on in-depth interviews with MR professors from universities in North America. Bridges' results show that professors try to incorporate quantitative and qualitative research into their syllabi. However, the increase in statistical analyses, which is a consequence of the development of IT, forces them to choose which aspects are critical. However, with the same number of hours for the subject, the time allotted to cover the content decreased.

In said work, Bridges found five critical points in the design of the MR university course, which corresponded to the stages of a market study: determining the purpose of the study, establishing the focus (qualitative or quantitative), designing the study, collecting and analyzing data, and, finally, presenting the results.

The author also brought a new debate: the interest in working with secondary data. Some professors focus on available information, which students must locate and analyze, whereas others feel it is more critical to cover primary research in MR courses.

Finally, in Bridges' study, professors agreed on the reluctance of MR students to approach statistics. Despite the general use of statistical software (SPSS) in universities,

professors were concerned because their students needed help understanding the necessary processes and, consequently, could not explain their results.

Nonis and Hudson [24] noted that MR students found it challenging to understand the contents of the course, given that it implied previous knowledge that, on occasion, must be added. It is an additional problem for professors because they must invest more time in covering the content of earlier courses. It leads to lower overall grades, a lack of satisfaction with the period, and changes in university majors. Bridges [21] studied a sample of 488 marketing course evaluations and concluded that the 48 MR courses were significantly worse than the rest.

Other authors have studied the professional perspective, that is, the relationship between teaching MR at the university level and the requirements needed to obtain a position as a market researcher.

For Kover [25], one origin of the problem was in academics' tendency to develop their MR syllabi to reflect their own needs and the skills they considered to be the most appropriate for the training. However, these choices may differ from companies' demands for a specialized job in the field.

Bellenger and Bernhardt [26] found that businesspeople considered it necessary to deepen quantitative analysis further and provide the course with a more practical perspective.

Bonoma [9] noted that one of the leading causes of conflict was the lack of interaction between professors and businesspeople in the sector, which resulted in academics not sharing their theories with professionals; consequently, these professionals did not understand them. There was also scarcely any communication in the opposite direction that pointed toward the industry's demands. It makes it even more essential to consider MR an eminently practical discipline.

In recent years, several studies [27] have shown similar findings and exemplified the gap between the opinions of both groups regarding the content, structure, and skills taught in the course. The study by Stern and Tseng [28] is worth noting. The purpose of the study was to further understand the perspectives of both groups on how to teach MR. To this end, the authors surveyed CEOs and professors in the US and Canada. Stern and Tseng found several interesting results:

- ignificant differences stemmed from professionals' particular interest in carrying out case studies and simulated research projects, whereas professors preferred lectures on theoretical issues.
- According to academics, significantly more coverage regarding scales and other data measurement techniques, questionnaire design, univariate analysis, and secondary information search was required. According to professionals, multivariate analysis and the ethical aspects related to research needed more attention.
- The most important statistical analysis techniques for academics were descriptive techniques, review of statistical concepts, chi-square tests, *t*-tests, and statistical correlation techniques. For professionals, essential techniques were descriptive statistics, assessment of statistical concepts, *t*-tests, statistical regression, and analysis of variance.

Segal and Hershberger [29] used a sample of 610 MR-related job offers to assess the skills, experience, and knowledge levels required by the industry. The authors obtained exciting conclusions, such as the following:

- Professional experience was found to be more important than the knowledge obtained from university training (94% vs. 71%).
- Skills were more significant than training (83% vs. 71%).
- Quantitative skills were much more important than qualitative skills (60% and 17%, respectively).
- The computer software knowledge needed was higher in Excel than SPSS (31% vs. 13%).
- The demand for a master's degree in MR was deficient (17% of the job offers), but it was higher than the need for a Statistics degree (12%).

Through a similar methodology of examining job offers, Benítez [30] studied the profile of the personal and professional skills required by MR companies. The study's results were similar to those obtained by Segal and Hershberger [29]. However, there were some quantitative differences: professional experience and university degree were at the top, with both having an equal weight (7.04%) and Excel skills being highlighted (4.23%).

Given the differences between professionals and academics in the MR field, both groups should establish a regular dialogue regarding the needs and depth needed for each technique, course topic, and method taught. As noted by Segal and Hershberger [29], it is more necessary in the MR field than in any other field to close the gap between academics and professionals, and for students to ultimately acquire the knowledge and skills needed for professional purposes during their time at university.

Finally, López [31], in a much more recent study, studied the opinions of students regarding their MR courses. The students valued all items positively (with a score above three on a Likert scale of one to five). However, teamwork was remarkably well-evaluated, as were the ability to apply the theories studied and the analysis of data and results.

Because of the problems found, some authors have proposed alternatives to the current design of the MR courses that are oriented towards trying to solve student weaknesses.

Nonis and Hudson [24] developed a proposal to improve the training given before starting the MR course. Specifically, they suggested creating a statistical basis by substituting the only statistical course available in BA degrees in the US (as it mainly occurs in Europe) with two other courses to be taught in the first two academic years. The first course would teach the fundamentals (Business Statistics), and the second one would focus on practical aspects (Applied Research Course) that include training in statistical software (SPSS).

Burns and Bush [32] proposed teaching the course using a staged approach. The objective would be to cover as many MR concepts as possible, but only for those students who had successfully passed a series of exams.

Kennett et al. [33] proposed an alternative method for evaluating students through a kind of MR portfolio. As students advanced through the syllabus, they would carry out projects related to each course theme. Moreover, this option would appeal to students who value experience and learning more than the traditional approaches.

In a much more recent study, Méndez and Ballina [34] analyzed the MR course programs in BA degrees in 76 Spanish universities, obtaining the following results:

- There was only one MR course per institution, with an average of 6.5 credits.
- The most frequent term used was MR (more than 40% of universities), followed by commercial investigation (20% of universities).
- The program had eight topics divided into four main areas.
- Practical classes were included, with case studies being more popular than classes on software (73% vs. 52%), and the use of IBM's SPSS software stood out.
- Continuous evaluation was used following two main mechanisms: individual assignments (66% of the cases) and group work (33% of the degrees).
- The two most referenced manuals were, in order: Malhotra et al. (2008), as seen in 23% of cases, and Huir et al. (2010), as seen in 10% of cases, in addition to others from national authors.
- All universities studied multivariate analysis techniques. Factor analysis (54% of the programs) and cluster analysis (39% of the cases) were the most prominent.

Finally, it should be noted that since 2018, Spanish universities have introduced the first official master's degree programs in data with the names "Big Data", "Data Analytics", and "Business Analytics". In addition, since the academic year 2020–2021, data engineering degrees have been developed in Spanish Polytechnic Universities. It resulted in a quantitative advancement in terms of MR development. Moreover, it was also a qualitative advancement given that, for the first time, university training in business information was moving from business and enterprise faculties to the schools of engineering.

### 3. Empirical Study—Academics vs. Professionals

The overall objective of this study was to evaluate the significant changes that have taken place and continue to take place in terms of the availability and types of data. In addition, developing new analytical techniques should encourage changes in the structure and content of the MR courses taught in Spanish universities. Different phases or subobjectives have this goal (Figure 1):

1. Is there "old" content that should be removed from current MR course programs?
2. Should MR at the university level include other content relevant to professional practice?
3. Should the "intensive" statistical–analytical content of current MR courses be modified?
4. What would the quantitative (credits) and qualitative structure of the "new" MR course be at the university level?

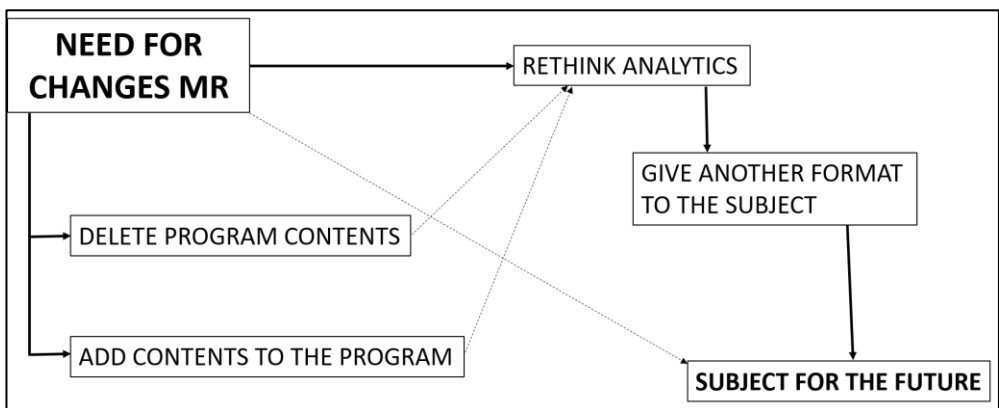

**Figure 1.** Outline of the objective and subobjectives of the work.

Some of these issues are closely connected. For example, the introduction of changes to the content of an MR program's subjects will have to produce transformations in the methods of analysis required for such content. In addition, the perception of the need for changes by the subjects studied will relate to the new structure necessary for the "new" MR subject.

To this end, and following the previous literature, a structured field study was designed in two interrelated parts: a survey completed by MR professionals from different universities across the world, and another survey conducted by professionals from the primary MR and data companies in Spain, many of which are multinational (Table 2).

**Table 2.** Technical sheet of the study.

| Sample unit | University professors of MR subjects | Active MR technicians |
|---|---|---|
| Census size | Universities from 30 countries around the world | 23 most prominent MR institutions in Spain |
| Sample size | 301 valid questionnaires | 304 valid questionnaires |
| Survey method | Self-administered (email survey) | |
| Study period | May and June 2021 | May and June 2022 |

The design of the questionnaire considered the work of Bridges [21,23] and Méndez and Ballina [3] carried out with MR university professors. Logically, the study omitted the items related to methodology, teaching, and evaluation of the subject because they were not convenient for the group of professionals. We worked with ten questions: three general questions, two questions about parts that are "unnecessary" in the program, two questions about the need to "add" ICT data, two questions about the relevance of multivariate statistical techniques, and an open question for additional input (Table 3).

**Table 3.** Questionary Structure.

| Block | Variables (Q) | Scales |
|---|---|---|
| About the aubject of marketing research | V1 = Needs Changes<br>V2 = Preferred Name<br>V3 = Number and Type of Credits | Lm<br>Lm<br>Ordinal |
| About parts to "remove" | V4 = Remove Basics<br>V5 = Remove IM Fonts<br>V6 = Remove SIM<br>V7 = Remove Design<br>V8 = Remove Qualitative T<br>V9 = Remove Survey<br>V10 = Remove Questionnaire<br>V11 = Remove Sampling<br>V12 = Remove Panels<br>V13 = Remove Experimentation<br>V14 = Remove Data and Databases<br>V15 = Remove Univariate Statistics<br>V16 = Remove Bivariate Statistics<br>V17 = Remove Multivariate Statistics<br>V18 = Remove Specific IM Applications<br>V19 = Remove Report | Lm |
| About ICTs to "add" | V20 = Add Big Data<br>V21 = Add CRM<br>V22 = Add Dashboards<br>V23 = Add Geolocation<br>V24 = Add Google Analytics<br>V25 = Add Artificial Intelligence<br>V26 = Add Virtual Reality Research<br>V27 = Add Metasearch Engines<br>V28 = Add Review Analysis<br>V29 = Add RRSS Monitoring<br>V30 = Add Observation by Sensors<br>V31 = Add ICT Panels<br>V32 = Add Neural T | Lm |
| About multivariate techniques | V33 = Do Component Factor Analysis<br>V34 = Do Factor Analysis Correspondences<br>V35 = Make a Hierarchical Cluster<br>V36 = Make a Nonhierarchical Cluster<br>V37 = Linear Regression<br>V38 = Multivariable Regression<br>V39 = Logarithmic Regression<br>V40 = Make Multi-ANOVA<br>V41 = Do Simple Discriminant Analysis<br>V42 = Multiple Discriminant Analysis<br>V43 = Do Conjoint Analysis<br>V44 = Multidimensional Scale<br>V45 = Make T Forecast | Lm |
| Final comment | V46 = Added Input on the Issue | Open Nominal |

Lm: Likert/metric scale.

In accordance with the previous literature, four main hypotheses were developed (Figure 2):

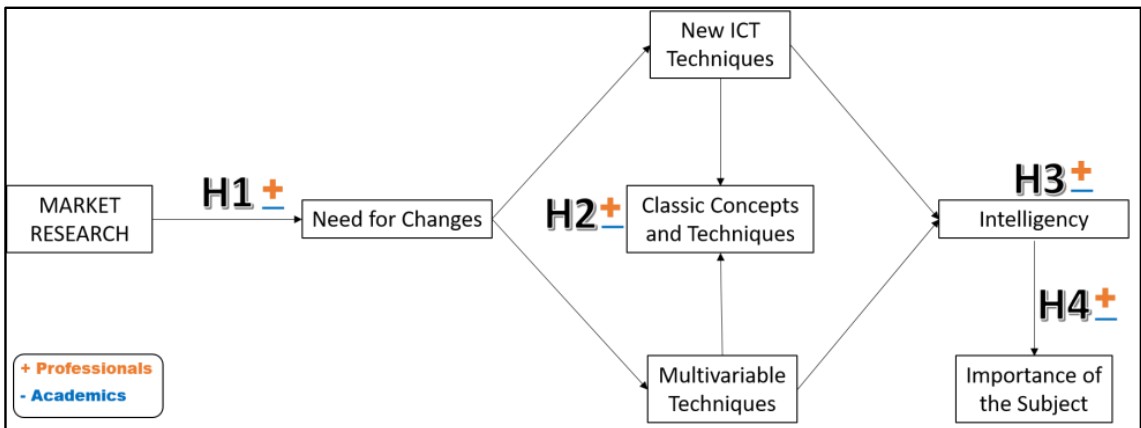

**Figure 2.** Hypothesis Scheme.

**H1.** *The need to change current MR course programs is positive for professionals and negative for academics.*

The classic works of Kover (1976) and Bellenger and Bernhardt (1977) support H1. They note that professors design MR programs according to their needs and preferences and even avoid interaction with MR companies (Bonoma, 1988).

**H2.** *Professionals, not academics, support removing concepts in current MR course programs.*

Univariate analysis of statistics develops H2A. Thus, Nonis and Hudson (1999), Burns and Bush (2010), and Kennett et al. (2010) advocate for the need to remove reiterative studies in BA degree courses.

Two different approaches have H2B: one direction of authors who support more qualitative research (Evans, 2002; Paas, 2019) versus another who call for the further development of multivariate analytics (Benítez, 2016; Bridges, 2020; Segal and Hershberger, 2006 and Stern and Tseng, 2002).

**H3.** *Professionals rather than academics propose the change toward the term "intelligence" instead of "research".*

**H4.** *The importance of the MR course, as measured in the number of credits, is increasing for professionals and remains the same for academics.*

Finally, H3 and H4 run parallel to each other. Early contributions by Malhotra and Peterson (2001), and, more recently, Ghorbani et al. (2022), are aligned, as they suggest adapting the term used to describe the field to the new trends in the professional sector, thus promoting the expansion of the content and, therefore, the weight of the MR university course.

## 4. Results

The data were processed using various comparative statistics. IBM SPSS v.27 was the software used.

First, the results presented in Table 4 are the final distribution of the sample regarding the professionals and academics and the distribution of the sample by major countries.

**Table 4.** Composition of the final sample.

| Type | Percentage |
|---|---|
| Professionals | 50.2 |
| Academics | 49.8 |

**Table 4.** *Cont.*

| Type | | Percentage |
|---|---|---|
| | Spain | 11 |
| | USA | 18 |
| | Australia | 12 |
| | Brazil | 7 |
| | Colombia | 6 |
| | United Kingdom | 5 |
| Country | Germany | 4 |
| | India | 4 |
| | France | 4 |
| | China | 4 |
| | Mexico | 3 |
| | Others | 22 |

The reliability of the Likert scale used for the 45 variables of the questionnaire (see Table 3) shows satisfactory Cronbach's alpha results above 0.0 (Table 5):

**Table 5.** Scale Validation.

| Statistical | Value |
|---|---|
| Cronbach's alpha | 0.824 |
| Cochran's Q test | 0.000 |
| Hotelling T-squared statistic | 0.000 |

First, regarding the H1 hypothesis, the analysis evaluates the mean differences in the value professionals and academics each give to the demand for change in MR programs. The *t*-test for independent samples indicates that there is a slightly higher, but significant, mean for the case of the professionals. The analysis considered three issues regarding changes to the program: removal, adding multivariate statistics, and adding ICT. The results of the *t*-tests showed significant differences between the three cases. However, whereas the mean for "adding aspects" (ICTs and multivariate statistics) was higher in the professional group, according to H1, this is not the case with the question of "removing aspects," where the academics presented a higher mean (Table 6).

**Table 6.** Mean difference *t*-test for independent samples.

| Academic vs. Professional | | Stocking | Standard Error Mean | Sig. (Bilateral) |
|---|---|---|---|---|
| Need for program changes | Academician | 2.870 | 0.0390 | 0.002 |
| | Professional | 3.040 | 0.0360 | |
| The average value of removing | Academician | 3.1497 | 0.03267 | 0.000 |
| | Professional | 2.4307 | 0.01968 | |
| The average value of multivariable | Academician | 3.6121 | 0.03474 | 0.000 |
| | Professional | 3.9739 | 0.03814 | |
| The average value of adding ICT | Academician | 3.4695 | 0.03535 | 0.000 |
| | Professional | 3.7831 | 0.02945 | |

To address H2, a deeper study into the different options to "remove aspects" from the program is needed. A simple discriminant analysis (SDA) was carried out in which the dependent variable was academics vs. professionals. Nine of the 16 options studied were significant (stepwise statistics and Wilks' lambda), which, by relating their coefficients to the centroids of each type of individual, indicates that (Table 7):

- Academics (centroid +) favored removing (in this order): qualitative techniques, specific applications of MR, univariate statistics, report writing, and experimentation.

- Professionals (centroid −) favored removing (in this order): sampling, panels, databases, and basic concepts.

**Table 7.** SDA over "remove" from the MR program.

| Standardized Canonical Discriminant Function Coefficients | Value |
|---|---|
| Remove Basics | −0.196 |
| Remove Qualitative T | 0.701 |
| Remove Sampling | −0.559 |
| Remove Panels | −0.245 |
| Remove Experimentation | 0.304 |
| Remove Data and Databases | −0.218 |
| Remove Univariate Statistics | 0.344 |
| Remove IM-Specific Applications | 0.649 |
| Remove Report | 0.325 |
| Functions in group centroids | |
| Academician | 1.845 |
| Professional | −1.827 |

After applying the SDA technique for the case of "adding" ICT techniques to the course program, using the same statistical process, seven significant options out of the 13 studied were identified (Table 8):

- Professionals (centroid +) considered it most relevant to add (in this order): big data, technology dashboards, and social media monitoring.
- Academics (centroid −) considered it most relevant to add (in this order): observation by sensors, metasearch engines, artificial intelligence, and CRM.

**Table 8.** SDA over "adding ICT" to the MR program.

| Standardized Canonical Discriminant Function Coefficients | Value |
|---|---|
| Add Big Data | 0.724 |
| Add CRM | −0.145 |
| Add Artificial Intelligence | −0.298 |
| Add Metasearch Engines | −0.298 |
| Add RRSS Monitoring | 0.420 |
| Add Observation by Sensors | −0.400 |
| Add ICT Panels | 0.478 |
| Functions in group centroids | |
| Academician | −1.088 |
| Professional | 1.077 |

Other ICTs proposed in an open question by the professionals were dynamic dashboards, audio matching, and bots (Table 9).

**Table 9.** Other ICT techniques to add to the program.

| Other | Percentage |
|---|---|
| Use of Excel | 9.2 |
| Sector-specific Techniques | 27.7 |
| Structural Relationships | 9.2 |
| HALO | 9.2 |
| Project Management (Agile) | 9.2 |
| Decision Trees | 18.5 |
| Bayesian Analysis | 9.2 |
| Algorithms | 7.7 |

The third SDA applies to the case of multivariate techniques. In addition, it used stepwise statistics and Wilks' lambda protocol. There were ten effective techniques out of the 13 considered, indicating that (Table 10):

- Professionals (centroids +) proposed adding (in this order): multiple discriminant analysis (MDA), linear regression, non-hierarchical clustering, and conjoint analysis.
- Academics (centroids −) proposed adding (in this order): multivariate regression, multi-ANOVA analysis, simple discriminant analysis (SDA), and multidimensional scaling (MS).

**Table 10.** SDA over "adding multivariate statistics" to the MR program.

| Standardized Canonical Discriminant Function Coefficients | Value |
|---|---|
| Make Nonhierarchical cluster | 0.189 |
| Linear Regression | 0.373 |
| Do Multivariate Regression | −0.896 |
| Do Logarithmic Regression | 0.206 |
| Make Multi-ANOVA | −0.441 |
| Do Simple Discriminant Analysis | −0.360 |
| Do Multiple Discriminant Analysis | 1.181 |
| Make Multidimensional Scale | −0.139 |
| Make T Forecast | 0.268 |
| Group centroid functions | |
| Academician | −1.554 |
| Professional | 1.539 |

Moreover, professionals proposed to incorporate another technique that mainly highlighted the case of sector-specific techniques and decision trees (Table 11).

**Table 11.** Other statistical skills to add to the MR program.

| Skills | Percentage |
|---|---|
| UX | 3.8 |
| Hybrid Methodologies | 2.5 |
| Audio Matching | 23.8 |
| Dynamic Dashboard | 34.7 |
| Boots | 19.2 |
| Programming R/Phython | 15.9 |

Concerning H3, the proposed future name for the MR course, another *t*-test studied the differences in means, which presented the following results (Table 12):

- Academics showed a significant preference for names tagged with the word "research" in the following order: "marketing research", "market research", and "applied research".
- Professionals preferred the words "intelligence" and "studies" in the following order: "market intelligence" and "market studies".

Concerning the term "business intelligence", there were no significant differences, although the values were lower than in the rest of the cases.

Finally, analysts performed a chi-square test in terms of H4 and regarding three outputs: the weight of the course, the format, and the number of credits. The test concluded that there were significant differences between the two types of sampling units (Table 13):

- Academics proposed to keep the MT course, as it is mandatory in the BA degree and worth six ECTS credits.
- Professionals would also add a new elective MR course worth three ECTS credits.

**Table 12.** Mean difference *t*-test for independent samples.

| Academic vs. Professional | | Stocking | Standard Error Mean | Sig. (Bilateral) |
|---|---|---|---|---|
| Market Research | Academician | 3.84 | 0.075 | 0.000 |
| | Professional | 3.13 | 0.063 | |
| Marketing Research | Academician | 4.08 | 0.050 | 0.000 |
| | Professional | 3.23 | 0.077 | |
| Research Applied to Business | Academician | 3.25 | 0.080 | 0.000 |
| | Professional | 3.63 | 0.074 | |
| Market Studies | Academician | 3.25 | 0.080 | 0.000 |
| | Professional | 3.63 | 0.074 | |
| Market Intelligence | Academician | 2.26 | 0.055 | 0.000 |
| | Professional | 3.94 | 0.075 | |
| Business Intelligence | Academician | 2.91 | 0.061 | 0.000 |
| | Professional | 3.02 | 0.085 | |

**Table 13.** Differences (Oi-Ei) chi-square test.

| | Academics | Professionals |
|---|---|---|
| Elective six credits | −1 | +1 |
| Compulsory six credits | +12 | −12 |
| +Elective 3 credits | −6 | +6 |
| +Elective 6 credits | 0 | 0 |
| Compulsory 12 credits | −3 | +3 |

In summary, the hypotheses to accept or reject are:

- H1: Accepted.
- H2A: Rejected.
- B Accepted.
- H3: Accepted.
- H4: Rejected.

## 5. Discussion

This study shows the necessity to provide students taking MR courses with more previous knowledge, which professors see as an opportunity to take more time to develop their academic programs.

However, professors suggest a further reduction of the current content. The debate regarding training in qualitative research is still open, and the ability to communicate with reports divides both interest groups.

Both professors and professionals recognize the introduction of complex statistical methodologies and new smart techniques, although the professionals are significantly more noteworthy in this case. The case of big data is critical, perhaps due to the effect that it has on the development of more master's and bachelor's degrees related to it that are unrelated to MR training.

In any case, the results do not suggest mutually exclusive positions; in fact, it is quite the contrary. One reason is that professionals continue to advocate for the term "market", and professors opt for "research". However, above all, this is because, whereas professors support a significant restructuring in the contents of mandatory MR courses, professionals propose an increase of content, which would allow training in new techniques and concepts through an additional elective course. As such, there is a proposal to work on aligning both perspectives in universities.

## 6. Conclusions

MR has experienced an evolutionary process parallel to marketing at the academic and company levels. However, we are now at a point of enormous expansion in all aspects

related to data management and even the use of MR to get closer to the general functions of business management.

This is due to the increased complexity and the significant momentum of change in today's markets. Moreover, it also can be explained by the constant search for precision in MR methodologies and techniques. Thanks to software development, the advances in the use of complex statistical techniques have combined with the result of qualitative research, neural network techniques, and the possibility of working with NoSQL data.

Today, smart technologies are driving a new revolution, mainly due to big data, artificial intelligence, and virtual reality developments. Universities are experiencing an increasing divide between teaching MR and developing new master's and bachelor's degrees in data engineering.

Several authors have noted the gap between the MR programs taught at the university level and the knowledge and skills that professionals and market research institutions need today. It is due to the significant difference in adaptation times with which universities, especially public universities, operate. The age of university professors, the search for comfort zones in their teaching practice, as well as the difficulties in recruiting technicians specialized in new technologies provide support and act as factors preventing a transition in the short term.

Three agents participate in MR training: professors, professionals, and students. Each agent has their objectives, which are only sometimes in line with the teaching–learning training that should characterize the teaching of MR courses in universities.

According to the referenced works, students show an increased interest in MR course content and related skills. However, they are concerned about difficulties arising from increased statistical knowledge demands.

It is necessary to have a balanced model built based on the needs of professionals. The distances could be wider than they are, though. There is enough consensus among the agents in maintaining a main structure of the MR subject very similar to the current one, both in fundamentals and concepts, with the addition of a new complementary subject being the way to approach the desired balance.

### 6.1. Academic and Business Implications

The first and most important aspect of this work is aligning the knowledge interests of professionals and companies with university-level training. The professional sector demands more complex statistical techniques and new technological tools at the university level. Moreover, the right path is not to create master's or bachelor's degrees in data engineering, which is distant from what MR should be. This approach is becoming increasingly popular in university training, which may result in data technicians needing knowledge that is useful or applicable to companies or management technicians who need help handling new data.

### 6.2. Limitations and Future Research

The main limitation of this work has been the restriction of working with surveys. In addition, it has been subject to the existence of a great diversity of MR programs in each of the universities considered. Academics must learn more about the new technologies and their MR possibilities. This issue affects the results obtained. Something similar occurs in the case of qualitative MR, which, although better known, has less use in practice in universities. It may have led to some bias in the results.

Meeting the challenges for a future research agenda is essential. As noted in the literature, the gap between practitioners' positions and academics remains and requires further research. The first tasks should relate to broadening the pool of stakeholders, first with students and second with new centers and university professors, linked to big data or data engineering degrees. From the perspective of the MR subject programs, it would be relevant to consider the existence of a hierarchy in smart technologies in their professional application, as well as in the confluence of qualitative and quantitative data of

the NoSQL type, which is hardly developed at the university level. Finally, there needs to be an essential fill in the knowledge gap in the study of the complementary tools that students should work on within the MR profession, whether related to the management of logic, statistics, or computer programming.

**Author Contributions:** Conceptualization, J.d.l.B.; Methodology, J.d.l.B.; Software, S.C.; Validation, J.d.l.B.; Formal analysis, J.d.l.B. and S.C.; Data curation, S.C. All authors have read and agreed to the published version of the manuscript.

**Funding:** This research received no external funding.

**Data Availability Statement:** https://drive.google.com/file/d/1J70pCzkJGvCG7k6O7xDjwhjD4 WWIsyBF/view?usp=share_link (accessed on 1 November 2022).

**Conflicts of Interest:** The authors declare no conflict of interest.

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
