# Peer review of "Teaching Marketing Research at the University Level—From Academic and Professional Perspectives"

_sustainability, doi:10.3390/su15021480_

Round 1

Reviewer 1 Report

The Authors have submitted an interesting paper involving possible changes to marketing research courses, by comparing the perceptions of professors and professionals. Nevertheless, there are multiple issues that must be addressed before being accepted. Here are a few comments per section of the paper: 

1.       Introduction:

a.       Overall, the section is good and provides a clear overview to the need for change in MR teaching in universities. I would recommend only minor modifications. For example:

o   The authors need to clarify the acronyms for SQL and NoSQL data. Every acronym must be explained the first time they are cited.

o   Please revise the writing of the section, to avoid misuses such as “it is important to highlight the importance of Neuromarketing a..”.

2.       The title of the section is written in Spanish (La evolución del concepto de Marketing Research). Please translate.

a.       The authors mention that in 1910 MR was first used in a US company. Which one? Be precise.

b.       One critical issue of the section is that most arguments on page 2 are not referenced. The description of the historical evolution of MR must be very well referenced with multiple sources.

c.        The description of BI is very generic. Please provide a more accurate definition.    

3.       Teaching Marketing Research at the university level

a.       In my view this section needs major modifications. I strongly recommend not replicating findings from previous studies in new tables, such as on Tables 1, 2, 3, 4, 5 and 6. These findings have been previously published by authors and replicating them in such style may infringe copyright issues. Thus, I suggest re-writing the entire section and paraphrasing the works and their findings. 

b.       Also, on Table 2 presents average scores higher than 5, when the table description says the items were measured from 1-5. Please review it.

c.        Writing: please review the verb tenses to be accurate and consistent. The verbs are often used in the present when they should be in the past. The inverse also happens. Thus, please review the entire paper in this regard for consistency.

4.       Empirical study. Academics vs. Professionals

a.       The rationale of Figure 1 should be improved. The authors have raised questions but have not explained the rationale being displayed in the figure. For example, what is the rationale of the direction of arrows in the figure? How do they relate to the research questions?

b.       Please provide a clearer explanation of how the variables from Table 8 were determined. Are there sources? If so, which ones? The authors only mentioned “adapted questionnaire for the professors”…  Please explain.

c.        To facilitate the reading, I suggest providing the explanation of each hypothesis after they have been mentioned. Currently all hypotheses are first listed and only after their explanations are given.

5.       Results

a.       The authors provide a Cronbach alpha score but have not mentioned to which factor(s) it is related to. Neither how many items were used. Also, the description says: “the database scales were analyzed”. If it was more than one scale, where are the other reliability results?

b.       For Table 10, please explain which type of t-test was used.

c.        The authors have described applying an ANOVA test (it is cited as Table 15). However why was an ANOVA used if the comparison involved only two groups? This must be justified, as an ANOVA in this case does not apply.  

Author Response

We wish to thank the reviewer for his thoughtful words about the paper, specifically for his rating of the paper as interesting. The changes (red color) are:

SQL and NoSQL data acronyms                     Explanations are included in parentheses.

Editorial....Neuromarketing                           The wording is modified to make it more understandable.

Title 2 in English                                             Apologies. Translator error. Already changed.

U.S. Company                                                 The Company name is included: CURTIS PUBLISHING

Page 2 is not referenced                                This is an important error, the references are now incorporated.

Description of the BI is very generic             An explanatory table (table 1) is incorporated to improve the explanation of the BI concept.

Do not reproduce the results of

previous studies

(Tables 1, 2, 3, 4, 5 and 6)                              Strongly agree with the reviewer. All tables are deleted.

Table 2 mean scores

above 5 are presented                                   Table deleted as above.

Review verb tenses                                         Thank you. Spelling was checked by the authors and then by a translator.

Improve justification of Figure 1                   A new paragraph is added to explain the arrows in Figure 1.

Sources of the variables in Table 8                The reference authors of the variables are already included (now Table 4).

Explain each hypothesis after                       The order of the text is changed, and following the reviewer's advice the explanation appears after each hypothesis.

Explain Cronbach's Alpha                               The information on the statistic is expanded (now Table 5).

Test used in Table 10                                     The name of the test is included in the title of the table: Mean Differences T-test (now table 6).

Error in the ANOVA test                                 Thank you. This is an important error. The test is a Chi-Squared test (now table 11).

Reviewer 2 Report

Attached ,  you  can  find  my  comments !

Author Response

Revise English language and style                 All text, changes included, is revised, first by the authors and then by a translation service.

Reviewer 3 Report

The article points out the need for alignment regarding Marketing Research training between professors, students, and professionals is becoming increasingly more urgent. This work is based on a double survey administered to professors and MR professionals in Spain to determine their proposals for adapting the format and content of the Marketing Research course to the current information needs of companies. The article, therefore, addresses an important problem facing academic communities. However, several changes need to be made to it.
1)    Line 60 is written in Spanish (not English)
2)    The article lacks a clearly marked purpose and research gap.
3)    I recommend adding a section in which you compile the main research from the academic literature. The literature review section is missing. The work has an introduction. However, I believe that it should also include an extensive section collecting the main review of the literature.
4)    The work presents results and conclusions. To complete the work, I recommend adding a section "The discussion". The discussion is very important to add to the work.
5)    Please provide additional research limitations and suggestions for future research.

Author Response

We wish to thank the reviewer for his thoughtful words about the work, specifically the idea of it being necessary work. The changes (red color) are:

Line 60 is written in Spanish                         Apologies. Translator's error. Already changed.

Improve the purpose and research gap        Thank you. The introduction contains changes to achieve this objective.

The literature review section is missing        To clarify the issue, Section 2 is called Literature Review. With two sections: the first on the literature on the evolution of market research, and the second on RM teaching at university.

Add a Discussion section                                Added as Section 5.

Provide the Incorporated Limitations           It is a new Section 6.2.

Round 2

Reviewer 1 Report

Dear authors,

Thank you for addressing many of the issues raised in the first round of review. Unfortunately, many issues still remain. There are still multiple English mistakes (e.g. . Pg. 3, "While BI is focus", Pg. 9 "Hypothesis Scheme (should be plural) ") and the paper still needs to be revised to improve its overall academic writing style and contribution.

In terms of structure, I would not advise including bulletpoints to discuss one's work, nor develop a literature review. I strongly suggest re-writing these sections.

Furthermore, much of the rationale and arguments still lack referencing to support itself (e.g. Pg. 4 ". This was a moment of independence for Marketing (???), however, there was still a very small number of courses, very disparate contents and denominations (be specific), and few compulsory courses").

On Table 5, it is still not clear what scale the Cronbach's alpha is testing.

The limitations and future reserch sections are very brief. The authors consider the constrainst of surveys as the main limitation. Please consider other additional limitations, which found on the paper. Also, the research agenda is very brief and is not linked with the existing literature.

Author Response

REVIEWER #1

English language

The document has undergone a double review: first, by the Grammarly Expert software, and second, by a native speaker reading.

Bullets

Have been converted to Tables

References (page 4)

Thank you. Indeed the references needed to be completed. We detect some numbering errors.

Scale Table 5

We add the type of scale that is the subject of the reliability study.

Limitations and Future Research

Thank you. We add a new paragraph for both questions. It clarifies the Limitations and, above all, the following steps for future work.

REVIEWER #2

Thank you very much for your review and final acceptance.

REVIEWER #3

Thank you very much for your review and final acceptance.

Round 3

Reviewer 1 Report

Thank you for addressing the changes.